# The Neuroimmune Axis and Its Therapeutic Potential for Primary Liver Cancer

**DOI:** 10.3390/ijms25116237

**Published:** 2024-06-05

**Authors:** Santosh K. Mandal, Poonam Yadav, Rahul A. Sheth

**Affiliations:** Department of Interventional Radiology, University of Texas MD Anderson Cancer Center, 1515 Holcombe Blvd., Unit 1471, Houston, TX 77030-4009, USA; skmandal@mdanderson.org (S.K.M.); pyadav1@mdanderson.org (P.Y.)

**Keywords:** liver cancer, cancer neuroscience, cancer immunity

## Abstract

The autonomic nervous system plays an integral role in motion and sensation as well as the physiologic function of visceral organs. The nervous system additionally plays a key role in primary liver diseases. Until recently, however, the impact of nerves on cancer development, progression, and metastasis has been unappreciated. This review highlights recent advances in understanding neuroanatomical networks within solid organs and their mechanistic influence on organ function, specifically in the liver and liver cancer. We discuss the interaction between the autonomic nervous system, including sympathetic and parasympathetic nerves, and the liver. We also examine how sympathetic innervation affects metabolic functions and diseases like nonalcoholic fatty liver disease (NAFLD). We also delve into the neurobiology of the liver, the interplay between cancer and nerves, and the neural regulation of the immune response. We emphasize the influence of the neuroimmune axis in cancer progression and the potential of targeted interventions like neurolysis to improve cancer treatment outcomes, especially for hepatocellular carcinoma (HCC).

## 1. Introduction

The role of the nervous system in regulating visceral organs is paramount. Recent advances in imaging and metabolic profiling have illuminated both the neuroanatomical networks within solid organs as well as their mechanistic impact on organ function [1]. Increasing evidence underlines the nervous system’s potent influence on the immune cells and tumor microenvironments within solid organs [2]. The nervous system regulates a broad spectrum of physiological and pathophysiological functions. It comprises the central nervous system (CNS) and the peripheral nervous system (PNS). The CNS includes the brain, brainstem, cerebellum, and spinal cord. It plays a crucial role in cancer growth and metastasis by releasing neurotransmitters and hormones. The PNS is divided into the sensory, motor, and autonomic systems and is further divided into parasympathetic and sympathetic branches. Similarly, the immune system is dynamic and adapts to a wide range of internal and environmental stimuli. However, persistent exposure to these stimuli can lead to an unregulated immune response, resulting in chronic inflammation and cancer through the recruitment of exhaustive innate, adaptive, and reprogrammed myeloid cells [3].

The immune and nervous systems are complex networks that play crucial roles in maintaining the body’s homeostasis, defending against pathogens, and overseeing physiological processes. They exhibit a remarkable degree of interaction and shared characteristics, such as receptor and ligand expression, allowing for efficient communication between the two. This bidirectional exchange is evident during embryonic development, homeostasis, and disease states [4]. The liver functions as a critical immune regulatory organ, maintaining a delicate balance under physiological conditions to prevent excessive immune responses against endogenous, dietary, and microbial antigens through complex mechanisms. However, this inherent immune tolerance also facilitates persistent liver damage from chronic infections, progressively leading to chronic hepatitis, cirrhosis, and, eventually, hepatocellular carcinoma (HCC). The liver’s immune system devises a unique environment that allows tumor cells to evade and contribute to the initiation and progression of HCC. A thorough understanding of the tumor microenvironment (TME) in HCC is essential for identifying regulatory targets to combat the development of resistance to immune checkpoint inhibitors (ICIs). The TME in HCC compromises cancer cells, immune cells, cytokines, and the extracellular matrix. It is characterized predominantly by the presence of immunosuppressive cells such as Kupffer cells, tumor-associated macrophages (TAMs), regulatory T cells (Tregs), and myeloid-derived suppressor cells (MDSCs), along with their signaling pathways [5,6]. A significant challenge in treating HCC lies in overcoming the immunosuppressive effects of the TME on the immune response. The expression of PD-1, a member of the CD28 family, is widespread among most immune cells within the TME, including myeloid-derived suppressor cells, activated T cells, Tregs, natural killer (NK) cells, dendritic cells (DCs), and monocytes. The interaction of PD-1 with its ligands, PD-L1 and PD-L2, which are upregulated in various cancers, including HCC, sends inhibitory signals to T cells, allowing tumor cells to escape immune surveillance [7].

In healthy conditions, liver sinusoids feature a regular endothelium with fenestrations and no basement membrane, with hepatic stellate cells in a quiescent state. However, in HCC, the endothelial lining thickens, losing its fenestrations through capillarization, forming a discontinuous basement membrane. Moreover, tumor cells may even form parts of the vessel wall. Once activated, hepatic stellate cells secrete vascular endothelial growth factor, among other angiogenic factors, resulting in structurally and functionally aberrant vessels [8].

Nerve fibers play a role in the complex neural–immune–vascular network within the tumor microenvironment (TME) of hepatocellular carcinoma (HCC). Intratumoral nerves within HCC significantly influence disease progression and patient outcomes [9]. Attributes such as nerve diameter, nerve density, and their spatial relationship to cancer cells—including distant interactions to perineural invasion (PNI)—are critical factors in cancer metastasis and survival. Larger nerve diameters and higher nerve densities correlate with poorer clinical outcomes across various cancers [10]. Researchers found that a higher expression of tyrosine hydroxylase (TH) and vesicular acetylcholine transporter (VAChT) in cancerous tissues, indicative of sympathetic and parasympathetic nerve density, was associated with vascular invasion, clinical stage, and lower survival rates, with TH levels also linked to disease recurrence [11].

Evidence from genetic, chemical, and surgical interventions targeting tumor-infiltrating nerves highlights their active roles in cancer initiation and progression. For instance, chemical sympathectomy using 6-OHDA targets peripheral sympathetic nerve endings, inducing cell self-oxidation without affecting neuron cell bodies or crossing the blood–brain barrier, leading to peripheral sympathectomy of the autonomic nervous system.

Understanding the role of nerve fibers within the TME is of utmost importance in cancer research. They contribute to pathological processes by secreting neurotransmitters and growth factors, promoting tumor development, and enhancing intratumoral innervation. This underscores a dynamic, bidirectional relationship between nerve fibers and cancer cells. Furthermore, nerve fibers collaborate with other TME components to facilitate metabolic support and coordinate vascular growth, ensuring the supply of essential nutrients and oxygen for tumor survival and expansion. This intricate interplay highlights the necessity of understanding and targeting the neural components within the TME as potential therapeutic strategies in HCC treatment [10]. We highlight these advancements with a specific focus on the role of neuroimmune interaction in liver cancer.

## 2. The Autonomic Nervous System and the Liver

The autonomic nervous system plays an integral role in motion, sensation, and the physiologic function of visceral organs [12]. It stimulates tissue growth, healing, and regeneration as a crucial component of the wound-healing response [13]. The peripheral nervous system comprises afferent somatosensory, efferent motor, and autonomic neurons. The autonomic nervous system is composed of two major parts: sympathetic and parasympathetic nerves [14]. These specialized nerves facilitate bodily functions such as the fight or flight response as well as maintaining proprioception [15]. Preganglionic fibers originating from thoracic segments within the spinal cord are responsible for connecting with peripheral autonomic nervous tissues within sympathetic chain ganglia. These fibers unite to form greater splanchnic nerve bundles, which subsequently synapse with postganglionic nerves located at the celiac ganglion [12]. Most neurons in the mouse celiac ganglion are tyrosine hydroxylase sympathetic and Neuropeptide Y-positive neurons. Interestingly, all the tyrosine-hydroxylase-positive neurons are also positive for vesicular monoamine transporter 2 (VMAT2) and the norepinephrine transporter (NET) [16]. Retrograde studies have demonstrated that celiac ganglia innervate both the liver and spleen. Parasympathetic nerves can be detected through vesicular acetylcholine transporter/vasoactive intestinal peptide staining [17]. These nerves originate from the dorsal motor nucleus of the vagus nerve (Figure 1).

Sympathetic innervation to the liver is a highly conserved feature across species, though the anatomic distribution of these nerve fibers differs [18]. In rodents, sympathetic nerves travel with the portal triads, encircling the portal vein branches similarly to hepatic lymphatic ducts [19] (Figure 2). In humans, on the other hand, sympathetic nerves travel along the hepatic sinusoids to the hepatic lobules [20]. Independent of their anatomic location, sympathetic signaling affects the metabolic functions of the liver, particularly its role in glucose metabolism. Sympathetic signaling to the liver drives elevations in blood glucose levels by increasing the conversion of stored glycogen to glucose [21]. Experimental studies have shown that decreasing a-adrenergic receptor signaling in the liver results in excess glycogen production and storage [22]. Furthermore, in an example of nerve–metabolism feedback, elevated insulin levels can decrease sympathetic signaling to decrease glucose release [23,24].

The nervous system additionally plays a key role in primary liver diseases. For example, excess sympathetic tone to the liver is a driver of nonalcoholic fatty liver disease (NAFLD) [25,26]. Interestingly, while an increase in sympathetic nerve fibers has been observed in early NAFLD, it appears that these fibers atrophy and are eliminated in chronic disease. In murine studies, sympathectomy effectively reversed steatosis independent of other risk factors such as diet and body weight [27,28], while excess sympathetic stimulation caused the opposite effect.

A type of volumetric imaging called immuno-enabled three-dimensional imaging of solvent-cleared organs (iDISCO) was used along with sheet fluorescence microscopy to investigate the arrangement of nerves and blood vessels in the liver of mice [27]. Noradrenergic nerve axons formed a network around the four branches in the left lateral liver lobe. The portal area and central veins exhibited a concentration of nerves [27]. Three-dimensional segmentation revealed that wider nerve fibers were primarily located around the artery and bile duct, while thinner fibers created an intricate network around the different components of the portal triad [27]. Different nerve types were identified through various markers such as tyrosine hydroxylase (TH) for sympathetic nerves, PGP9.5 for general nerves, vesicular acetylcholine transporter (VAChT) for parasympathetic nerves, and calcitonin-gene-related peptide (CGRP) for primary sensory nerves [29]. Additionally, the authors observed that fibers penetrated the liver tissue and noticed minimal staining after treatment with the neurotoxin 6-hydroxydopamine, which gradually recovered over time [27]. Examination of CGRP staining revealed distributed nerve fibers surrounding major portal branches without noticeable differences in steatotic/steatohepatitis samples compared with control samples [27].

## 3. The Interplay between Cancer and Nerves

Until recently, the impact of nerves on cancer development, progression, and metastasis has been largely unknown. In a seminal paper, Magnon et al. [30] showed that prostate gland denervation inhibits prostate cancer progression. Since then, similar findings have been reported across the cancer spectrum [13], including in HCC [11,31].

Mechanistic studies have begun to unveil the biological foundation for this interplay between nerves and cancer (Figure 3). The preponderance of data indicates that nerves influence tumors by manipulating stromal cells in the tumor microenvironment. For example, sympathetic signaling stimulates tumor angiogenesis by releasing norepinephrine into the tumor microenvironment, activating endothelial cells [32]. Immune cells are a cornerstone for this interplay [24]. Sympathetic innervation promotes metastasis development through increased macrophage infiltration [33,34]. b-adrenergic signaling from sympathetic nerves increases the production of interleukin-6 (IL-6), a potent immunosuppressive and oncogenic cytokine [35]. Adrenergic signaling activates myeloid-derived suppressor cells [28]. Increased neural tissue is also associated with increased tumoral checkpoint expression and T-cell anergy [30]. Given the growing body of evidence underlying the significant role of the neuroimmune axis in cancer progression, several retrospective studies have been conducted to investigate the impact of sympathetic signaling modulators such as b-blockers on cancer outcomes. One such study in pancreatic cancer found an improvement in overall survival [36], with multiple prospective studies ongoing.

From a broader perspective, cancer patients often experience heightened and persistent levels of chronic stress [38]. In animal studies, persistently elevated stress levels were found to lead to an overdrive of the stress response pathways, chiefly involving the hypothalamic–pituitary–adrenal (HPA) axis and the sympathetic nervous system (SNS) [39,40]. This over-activation can result in the dysregulated release of stress hormones such as cortisol, norepinephrine, and epinephrine, which may, in turn, favor the tumor microenvironment [41]. Furthermore, these stress levels can affect the growth and biodistribution of sympathetic nerve fibers that infiltrate the tumor site. The catecholamines produced by these nerves bind to adrenergic receptors located on immune and cancer cells within the tumor. This interaction potentially favors tumor progression [42]. A significant loss of sympathetic nerve fiber density in prostate cancer [30], breast cancer [43], HCC [11,30], and head and neck cancer [44] is associated with good clinical outcomes. In contrast, a loss of sympathetic nerve fiber density and a high level of expression of β-adrenergic receptors in tumor tissue are associated with a bad clinical prognosis in human gastric [45] and colorectal cancers [46]. However, one study in human colon cancer suggested that sympathetic nerve fiber density and a high level of expression of β-adrenergic receptors are positive regulators of tumor grade, size, invasion, and lymph node metastasis [47,48].

## 4. Neural Regulation of the Immune System

Adrenergic receptors (ARs) are proteins found in cell membranes. They belong to the rhodopsin family/class A of guanine nucleotide-binding protein-coupled receptors (GPCRs). These receptors can be categorized into two groups, α and β, based on their structure, properties, and how they transmit signals [49]. G proteins are stimulatory receptors, whereas Gi proteins are inhibitory. Β-adrenergic receptors use G proteins to enhance the activity of cyclase, which leads to increased AMP (cAMP) levels [50]. This elevation in cAMP stimulates protein kinase A (PKA) and other downstream transcription factors. On the other hand, α2 Adrenergic receptors utilize Gi proteins for signaling [51]. Adrenergic receptors are expressed on immune cells within the tumor microenvironment. Adrenergic receptors (ARs) are expressed in both the innate and adaptive immune cells [52]. The α-AR and β-AR subtypes are expressed on innate cells, which include monocytes, macrophages, and dendritic cells (DCs) [52]. β2-AR is predominantly expressed in natural killer (NK) cells, a part of the innate immune system, and adaptive immune cells like B-cells, cytotoxic T-cells (Tc cells), and helper T-cells (Th cells) [53].

The autonomic nervous system influences the innate and adaptive immune response to tumors at multiple intersections. For example, catecholamine signaling can influence the first step in the cancer immunity cycle [54] by impacting the function of antigen-presenting cells. The main function of dendritic cells (DCs) is to present antigens to cytotoxic CD8 cells. However, this function is compromised when epidermal Langerhans cells are treated with catecholamines [55]. Moreover, activating β-2 adrenergic receptor (β2-AR) signaling diminishes the cross-presentation of proteins by DCs. This leads to a reduction in the proliferation of CD8+ T cells and the production of IL-2 [56]. The activation of the β2-AR reconfigures CD40 signaling in DCs directly by suppressing the phosphorylation of IκBα and indirectly by increasing the levels of the phosphorylated cAMP response element-binding protein (pCREB) [32].

Furthermore, the autonomic nervous system can modulate the tumor microenvironment by influencing myeloid-derived suppressor cells [57]. Myeloid-derived suppressor cells (MDSCs) are pathologically activated immature myeloid cells exhibiting unique phenotypic and morphologic characteristics and subdued phagocytic and immune-suppressive functions [58]. In tumor-bearing mice, MDSCs are characterized as expressing CD11b and GR-1, with GR-1 further subdivided into Ly6C and Ly6G isoforms. In mice free from tumors, CD11b+/GR1+ MDSCs constitute less than 3% of nucleated splenocytes. However, in tumor-bearing mice, this proportion dramatically rises to over 20% [59]. Both in humans with cancer and in tumor-bearing mice, an expansion of MDSCs has been noted [60]. The concentration of MDSCs in circulation and at the tumor site tends to correlate positively with tumor size but negatively with the response to antitumor treatment and overall survival [58]. MDSCs employ a variety of strategies to hinder immune responses. One of these strategies involves the release of immune-suppressing cytokines, such as TGF-β and IL-10, which directly impair T-cell function [61]. Chemical sympathectomy using 6-OHDA in mice produced a decrease in lung metastasis by attenuating the recruitment of myeloid-derived suppressor cells. Chronic restraint stress accelerates hepatocellular carcinoma growth by directing splenic myeloid cells to tumor sites through the activation of β-adrenergic signaling. The interaction between CXCR2 and the CXCL2–CXCL3 axis plays a crucial role in the stress-induced recruitment of myeloid cells into tumor tissues. These cytokines also aid in creating and preserving regulatory T cells (Tregs), which are highly immunosuppressive [61,62]. Additionally, MDSCs frequently overexpress programmed cell death ligand 1 (PD-L1), which robustly suppresses T cells and NK cells expressing the corresponding receptor (PD-1) [63]. By blocking β2-AR signaling via β-AR inhibitors, curbing the function of norepinephrine-releasing nerves, or deleting the β2-AR, the accumulation and suppressive functions of MDSCs can be reduced [64]. This is associated with an enhancement in the effectiveness of the antitumor immune response and a subsequent reduction in tumor growth [64]. MDSCs that accumulate in the TME form a barrier to disrupt antigen presentation by DCs to NK cells and T cells and release substances like TGF-β and IL-10, which increase the number of Treg cells. MDSCs directly hamper NK cell function using TGF-β in mice liver cancer models. MDSCs further weaken the immune system using the enzyme Arg-1 to suppress T cell receptors and activity [65,66]. Some neurochemicals, e.g., catecholamines, are produced under normal conditions, which helps with myeloid cell maturation through alpha-AR interaction. However, in pathological conditions, many more catecholamines are secreted to inhibit myeloid cell maturation, resulting in myeloid-derived suppressor cells through beta-AR interactions. This process of regulation of MDSCs by the ANS is crucial for MDSC formation and accumulation at pathological sites such as tumors [67,68]. Catecholamines can also re-polarize M1 to M2 macrophages, which leads to the synthesis and release of IL-10, an immunosuppressive cytokine for the TME. Decreasing catecholamine levels can also shift the immunosuppressive microenvironment by decreasing myeloid-derived suppressor cell (MDSC) recruitment, facilitating dendritic cell (DC) activation, and increasing the antitumor immune response via the impact on multiple leukocytes [68].

Furthermore, α2-AR agonists enhance antitumor immune responses by diminishing the presence of MDSCs within the tumor microenvironment. This activation leads to the recruitment of CD4+ and CD8+ T lymphocytes within the tumor microenvironment [69].

Sympathetic neural signaling can also directly influence lymphocyte function [70]. CD8 effector T cells are the principal cell-killing method in an adaptive immune response. When these lymphocytes migrate to a tumor, they are called tumor-infiltrating lymphocytes (TILs). CD4+ regulatory T cells (Tregs) are immunosuppressive cells in the tumor microenvironment [71,72]. An elevated ratio of CD8+ T cells to CD4+ Tregs in the tumor microenvironment suggests a more effective antitumor immune response [73]. Programmed death-1 (PD-1) is a checkpoint receptor expressed on the surface of T cells, including tumor-infiltrating lymphocytes (TILs) [74]. The binding of PD-L overexpressed on cancer cells to the PD-1 receptor on T cells diminishes the immune response against cancer cells. PD-L1 expression on a large number of tumor-associated nerves attenuates the antitumor response by decreasing the number of CD8+ tumor-associated lymphocytes in prostate cancer [75]. Beta-adrenergic receptors (ADRBs) are a type of cell-surface receptor that responds to catecholamines, like norepinephrine and epinephrine [76]. They are expressed in all cells but are overexpressed in multiple cancer types. Importantly, both cancer cells and immune cells within the TME can express ADRBs, which means they can respond to catecholamines in ways that could influence cancer progression [77]. β-blockers block the action of catecholamines on β-ARs. In a preclinical study, Bucsek et al. [78] found that the β-blocker propranolol significantly decreased the tumor growth rate and prevented tumor re-implantation, suggesting an influence on the adaptive immune memory. This effect was abrogated in immunosuppressed mice, suggesting that the effect is mediated by the immune system. The intratumoral T lymphocyte population was enriched with CD8 cells relative to Tregs in mice treated with propranolol relative to the control [78]. Moreover, the addition of propranolol significantly improved the efficacy of anti-PD-1 immune checkpoint immunotherapy [79].

The converse has also been noted in preclinical studies. Kamiya et al. [43] applied genetic engineering methods to persistently activate sodium channels within adrenergic nerves found inside tumors. The authors found that this manipulation led to the accelerated growth of murine breast cancers. This sustained release of NE within tumor-influenced immune checkpoint proteins, such as PD-1, PD-L1, and FOXP3, is known to greatly inhibit the body’s antitumor immune actions [80]. Sympathetic denervation abrogated the oncogenic effect. The authors also studied samples from patients and found that increased intratumoral nerve density was associated with worse clinical outcomes and increased expression of immune checkpoint markers. Likewise, Geng et al. [81] studied the influence of sympathetic signaling in lung cancer patients. The authors found that patients resistant to anti-PD-1 checkpoint inhibition therapy have increased plasma levels of norepinephrine (NE). This NE-driven resistance to anti-PD-1 therapy can affect the secretion of C-X-C Motif Chemokine Ligand 9 (CXCL9) and adenosine (ADO) in tumor cells, thereby inhibiting chemotaxis and the function of CD8 T cells. This modulation is believed to operate via the WNT7A–β-catenin signaling pathway [82].

## 5. The Autonomic Nervous System and Liver Cancer

HCC is one of the most lethal cancers in the world and is the fastest-growing cause of cancer deaths in the United States [13]. As opposed to many other malignancies, progress on new therapies for this cancer has been stymied by intrinsic tumor resistance. Indeed, the approval of sorafenib in 2007 was followed by a decade of negative Phase III clinical trials [83]. Thankfully, this disheartening succession of failures was recently broken by the evaluation of immunotherapies for HCC. The CheckMate 040 trial found an improvement in outcomes with the use of the immune checkpoint inhibitor nivolumab in the second-line setting [23]. Unfortunately, though, the objective response rate was low, with only 20% of patients responding in the dose-expansion cohort. In addition to intrinsic resistance to conventional chemotherapies, HCC is also highly resistant to immunotherapies due to its immunosuppressive tumor microenvironment, and one contributing factor may be the rich innervation to these tumors [84].

The significance of the neuroimmune axis in HCC lies in the important role sympathetic nerve fibers play in promoting tumor growth by modulating the tumor microenvironment through neurotransmitter release [85]. An intricate network consisting of both sympathetic and parasympathetic neurons exists within liver tissues as well as primary liver cancer. Notably, hepatocellular carcinoma (HCC) manifests a high density of nerve infiltration within the tumor [11]. Such occurrences appear to be linked to increased tumorous progression rates, shifts in the tumor microenvironment, and heightened incidence levels for adverse patient outcomes. The authors found that nerve fiber expression within the tumors was significantly correlated with alpha-fetoprotein (AFP) serum level, clinical stage, and cancer recurrence. Prominent levels of nerve markers were associated with poorer survival [86].

## 6. Mechanistic Underpinnings of the Neuroimmune Axis

The tumor microenvironment is an intricate network of various components, including cancer cells, immune cells, stromal cells, blood vessels, the extracellular matrix, and nerves. Among these, nerves are considered to be one of the most significant pathological constituents within this milieu [87]. Tumors secrete substances known as neurotrophic factors to promote nerve formation within the tumor. These include nerve growth factor (NGF), brain-derived neurotrophic factor (BDNF), neurotrophin-3 (NT-3), and NT-4/5. Neutrophins bind to neurotrophic tyrosine kinase receptors (NTRKs), including TRKA, TRKB, and TRKC, and the p75 neurotrophin receptor on nerve terminals [88]. In turn, activated nerve terminals promote the distribution and growth of nerve fibers and release neuroactive molecules, including neurotransmitters. Neurotransmitters like norepinephrine, when bound to adrenergic receptors on cancer and immune cells, foster tumor proliferation and metastasis [89].

Systemic inflammatory signaling pathways also connect cancer, the immune system, and nerves. Inflammatory cells are closely associated with both somatic and autonomic nerves through prominent inflammatory mediators, such as Prostaglandin E2 (PGE2), and neuropeptides, including Neuropeptide Y (NPY) and Corticotropin-Releasing Factor (CRF), at sympathetic nerve endings as a reaction to adrenergic stimulation, leading to neurogenic inflammation. This mechanism plays a pivotal role in promoting tissue repair and recuperation. However, if this biochemical mechanism is not properly regulated, it may lead to chronic inflammation and tissue damage.

## 7. Neuromodulation and Its Potential as an Anti-Cancer Intervention

Targeted neurolysis interventions are routinely performed in standard clinical practice, though principally for pain palliation [90]. A common indication for this procedure is to ameliorate pain symptoms for pancreatic cancer patients. Pain in this setting arises from either direct neural invasion by tumor cells and/or stimulation of splanchnic afferent nerve fibers. Splanchnic nerves are autonomic nervous system components that carry both sympathetic and sensory visceral nerve fibers. Both efferent and afferent splanchnic nerve fibers pass through the celiac plexus. Thus, celiac plexus neurolysis interventions improve cancer-associated pain by abrogating the sensory component of the splanchnic nerves. In addition, though, these interventions also suppress sympathetic signaling to visceral organs by eliminating the sympathetic nerve component from splanchnic nerves.

This ramification is well-known to those who perform such interventions as it can manifest in post-procedural complications such as diarrhea, which develops due to unopposed parasympathetic innervation following denervation of the sympathetic fibers.

Percutaneous neurolysis can be performed through either chemical or thermal ablation. The most common medications used for the former are phenol and absolute alcohol. When exposed to these agents, nerves undergo Wallerian degeneration as well as coagulative necrosis [90]. An iodinated contrast agent is often mixed with the neurolytic agent to visualize the distribution of the injected therapeutic [90]. Thermal neurolysis can be performed with either heat- or cold-based approaches [91]. These approaches are particularly helpful when a tumor invades the celiac plexus. Neurolysis procedures are performed under image guidance, typically either fluoroscopy or computed tomography. These interventions are safe but can result in several known complications beyond procedural risks, such as bleeding and off-target ablation. Unopposed parasympathetic tone following ablation of sympathetic signaling can lead to orthostatic hypotension. To prevent this, patients are often bloused with fluids before the procedure. The most common adverse effect of this procedure is diarrhea [92]. A study by Mary Y. Tadros and Remon Zaher Elia examined the effects of percutaneous ultrasound-guided celiac plexus neurolysis for pain management in 21 patients with advanced upper abdominal cancer. The patients underwent initial screening tests followed by the neurolysis procedure, which was performed by injecting 50–100% alcohol or 10% phenol under ultrasound guidance to ablate the celiac plexus. Pain levels were assessed using a visual analog scale (VAS) at multiple time points after the neurolysis. The results demonstrated that celiac plexus neurolysis was highly effective for achieving substantial pain relief in these patients [93]. Patients should be offered it as part of a multidisciplinary approach to managing severe, persistent abdominal pain [94].

Given the potent role that the neuroimmune axis plays in cancer progression, one promising approach to overcoming the limitations of immunotherapy for HCC is targeted abrogation of sympathetic signaling to the tumors by neurolysis. While both efficacy and toxicity limit the sustained suppression of sympathetic signaling via systemic therapies, local neurolysis interventions can have a profound impact on intratumoral nerve density and the attendant ramifications for tumor immunity. Thus, compelling data suggest that this standard-of-care intervention may hold the key to unlocking immunotherapy’s potential for HCC [85,95,96].

While the immunologic ramifications of splanchnic nerve neurolysis are unknown, there are compelling data suggesting that pharmacologic interventions suppress sympathetic signaling. Hiller et al. [97] conducted a randomized trial of the b-blocker propranolol versus a placebo for patients undergoing breast cancer surgery. Patients initiated propranolol therapy for 7 days before tumor resection. Tissue analysis of the resected tumor revealed significant changes in several signal transduction pathways for patients who received propranolol versus the placebo, including downregulation of Snail/Slug, NF-kB, and AP-1. Likewise, there was an increase in CD68+ macrophage and CD8+ T-cell infiltration into the tumors for patients who exhibited physiologic responses to propranolol, such as decreased blood pressure and heart rate. Thus, this “window of opportunity” trial illustrated the profound effect of even short-course sympathetic tone reduction on intrinsic tumor transcription and immune cell infiltration. Hopson et al. [98] conducted a Phase II study also looking at propranolol in the neoadjuvant setting for women with breast cancer. Propranolol was added to their neoadjuvant chemotherapy regimen. A total of 10 patients were enrolled. Most patients could tolerate the target propranolol dose, highlighting the feasibility of adding this medication in this clinical setting.

Sympathetic tone blockade has also been evaluated for its impact on metastases in the preclinical setting. Chang et al. [99] evaluated the addition of propranolol to doxorubicin in a murine model of triple-negative breast cancer (TNBC). Previous studies have shown that TNBC is richly innervated [100] and that, analogous to HCC, increased intratumoral nerve density is associated with poorer clinical outcomes [43]. To further support the clinical significance of b-adrenergic blockade, the authors retrospectively reviewed outcomes for breast cancer patients with non-metastatic TNBC. They found that even though patients receiving b-blockers were significantly older, there was still a significant decrease in the risk of metastasis and a significant improvement in cancer-specific survival. The authors further investigated the class of chemotherapies that combined most effectively with b-blockade and found that the metastasis-suppression effect was only seen in patients receiving anthracycline-containing chemotherapy regimens. To explore the mechanistic underpinnings of the interplay between anthracyclines and b-blockers, the authors investigated this treatment combination in a TNBC mouse model that develops spontaneous metastases. The preclinical results were concordant with their retrospective clinical findings in that propranolol in combination with doxorubicin decreased the rate of development of remote metastases compared with doxorubicin alone. On the other hand, the addition of propranolol did not decrease the growth rate of established lung metastases. Additionally, propranolol alone did not have a significant anti-cancer effect. Mechanistically, doxorubicin increased TH+ nerve staining within the tumor compared with untreated tumors. An incremental effect was not observed in non-anthracycline chemotherapies such as paclitaxel. Likewise, the combination of propranolol with these non-anthracycline drugs did not affect the development of metastases in the mouse model.

## 8. Parasympathetic Signaling—Friend or Foe?

The sympathetic nervous system has been identified as the principal driver of the oncogenic and immunosuppressive effects of the nervous system, as evidenced in various cancers such as breast cancer [43], pancreatic cancer [36,101], and HCC [11]. Conversely, the increased parasympathetic tone is protective against this cancer. This dichotomy, however, may not hold depending on the cancer type. That is, for some malignancies, it may be that parasympathetic signaling drives tumor progression. Gastric cancer is one such example [102,103]. Vagotomy, or the surgical or chemical denervation of the vagus nerve, is performed to alleviate the symptoms of gastric ulcers. Preclinical studies demonstrated that vagotomy suppressed tumor formation in a mouse model of gastric cancer [103]. A similar tumor-suppressive effect was observed in a genetically engineered mouse model of intestinal cancer [104]. Concordantly, cholinergic nerve density increases as a function of tumor progression [102]. Adrenergic signaling, though, likely also has a tumor-promoting effect in gastric cancer, an effect that can be abrogated through b-blockers [105].

Mechanistically, it has long been known that the gastrointestinal tract is richly innervated. While the parasympathetic and sympathetic nerves to the gut have their cell bodies outside the bowel, an enteric nervous system (ENS) with cell bodies located within the bowel walls also exists. During the initial development of gastric cancer, the density of cholinergic nerve fibers increases [102]. Acetylcholine released by the nerve endings activates the EGFR–ERK–AKT signaling axis and the Wnt signaling pathway, thus potentiating tumorigenesis [106,107]. Acetylcholine binding also increases the expression of matrix metalloproteinases, thus facilitating cancer cell migration [108,109]. Similarly, sympathetic signaling through the α and β2-AR can activate the EGFR–MEK–ERK pathway, also driving tumor progression [105]. Clinical studies have shown that increased levels of β2-AR present a worse outcome in colorectal cancer [110].

Nerve signaling also impacts the tumor immune microenvironment in gastrointestinal cancers. However, unlike other malignancies, gastrointestinal cancers do not exhibit an apparent dichotomy between immunosuppressive and immunostimulatory parasympathetic systems. In a fascinating preprint by Bauer et al. [111], the authors present data indicating that parasympathetic signaling can affect HCC growth by influencing the gut microbiome. The authors performed hepatic vagotomy and found that this intervention, targeting parasympathetic signaling to the liver, decreased HCC tumor growth significantly. Conversely, pharmacologic stimulation of the parasympathetic tone led to increased tumor growth. Immune profiling revealed a regulatory effect on CD8+ T cells by the parasympathetic nervous system. Since vagal tone influences the function of the gastrointestinal system, the authors postulated that the gut microbiome might play a role in the parasympathetic nervous system’s influence on the liver and HCC. They found that vagal tone and the presence of liver tumors influence the composition of the gut microbiome, which in turn influences hepatic immune tolerance.

## 9. Conclusions

The recent advancements and insights presented in this review highlight the interplay between the nervous system and the immune response in the context of primary liver cancer, particularly hepatocellular carcinoma (HCC) [112]. Dysregulation within the sympathetic nervous system can significantly lead to the development of liver cancer. Studies have elucidated the pivotal role of the sympathetic neuroimmune axis in cancer progression and metastasis, impacting both tumor angiogenesis and the behavior of immune cells and facilitating processes such as tumor growth, immune evasion, and metastasis through various cellular mechanisms [112,113]. The substantial presence of nerve fibers within both the tumor and stroma of hepatocellular carcinoma (HCC), along with an immunosuppressive microenvironment, presents a barrier to effective therapy. The neural input must be assessed because the degree of innervation depends on the tumor type and size. Unraveling the peripheral nerve network through molecular profiling of nerve fibers may hold promise for neuro-targeting cancer therapy. Direct and indirect networks of nerves and immune cells inside the tumor are yet to be fully understood in order to target abnormal nerve function within the tumor as the effective treatment strategy for cancer. Various preclinical pharmacological studies used the combined approach of beta-blockers and anti-checkpoint point inhibitors as a new treatment for cancer patients. However, it is important to note that β-blockers can have significant side effects, such as heart problems, low blood sugar, breathing difficulties, and neurological reactions like depression and fatigue. Developing a new viral tool to map and trace the specific group of aberrant nerves associated with cancer and immune cells is a promising avenue for manipulating nerves with emerging electroceuticals. Electroceuticals could help address this issue by providing less-invasive tools to control nerve signaling precisely. This can help inhibit the nerves that promote cancer growth while stimulating those that slow its progression. We can control how nerves communicate in the tumor area using different techniques like surgery, chemical ablation, and genetics. This could be a hopeful and effective way, in combination with an anti-checkpoint point inhibitor, to treat cancer. Consequently, targeting the modulation of the neuroimmune axis presents a promising therapeutic strategy to augment the efficacy of immunotherapy in treating HCC.

## Figures and Tables

**Figure 1 ijms-25-06237-f001:**
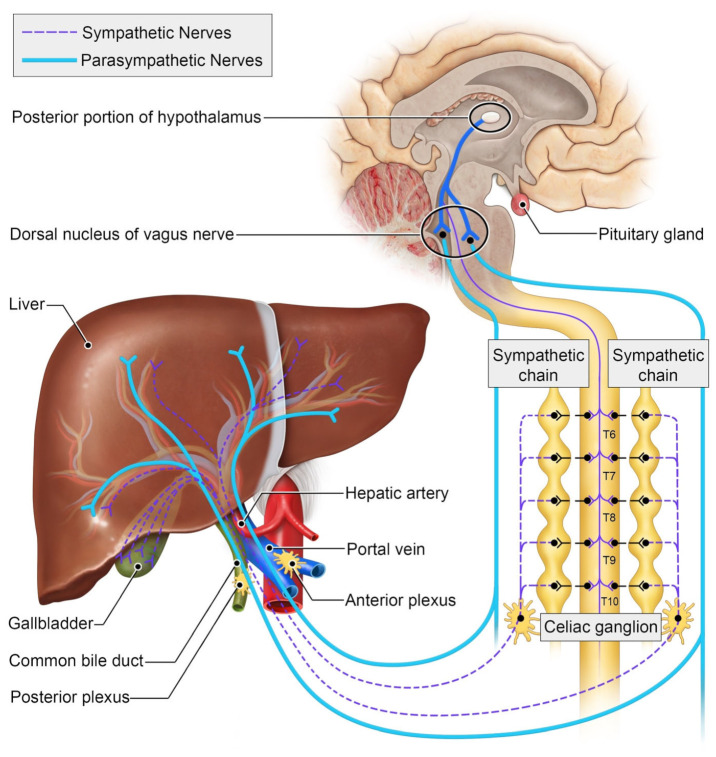
Schematic of the autonomic innervation of the liver.

**Figure 2 ijms-25-06237-f002:**
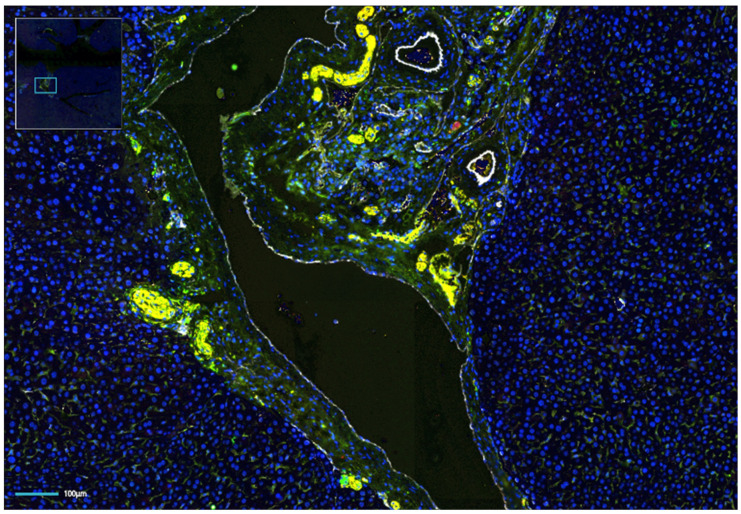
Multi-fluorescence image of an orthotopic hepatocellular carcinoma (HCC) in a syngeneic mouse liver. Nerve staining shows an abundance of intratumoral nerves. Green, TH; yellow, NF-H; red, PGP9.5; white, CD31; blue, DAPI.

**Figure 3 ijms-25-06237-f003:**
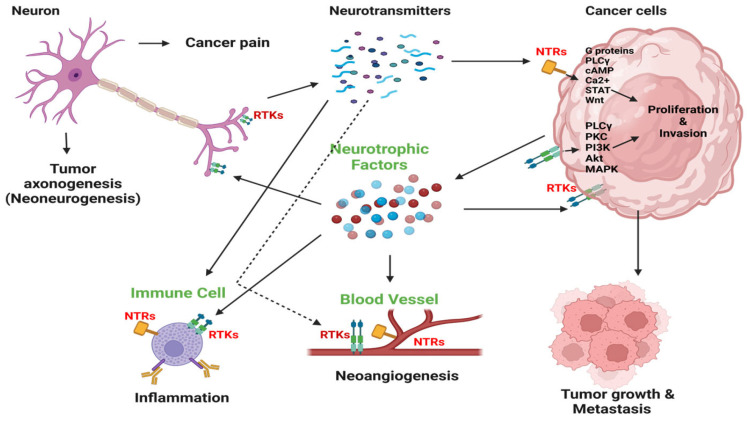
Nerve–cell communication within tumors. Nerves infiltrate the tumor microenvironment and stimulate cancer cell growth and metastasis by releasing neurotransmitters like catecholamines, acetylcholine, and neuropeptides. This initiates signaling pathways for cancer cell growth and invasion via neurotrophin receptors (NTRs). Cancer cells release neurotrophic growth factors such as NGF, promoting nerve infiltration into the tumor, also known as axonogenesis or neo-neurogenesis, and also receptor tyrosine kinase (RTK)-mediated autocrine stimulation. This reciprocal interaction fuels tumor development and affects the tumor microenvironment, leading to inflammation and angiogenesis. Cancer-induced pain may also result from tumor innervation. PLCγ, phospholipase C γ; cAMP, cyclic adenosine monophosphate; PKC, protein kinase C. Adapted from Jonling et al. [37].

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
