# Peer review of "The Neuroimmune Axis and Its Therapeutic Potential for Primary Liver Cancer"

_ijms, 2024, doi:10.3390/ijms25116237_

Round 1
Reviewer 1 Report
Comments and Suggestions for Authors
The article submitted by the authors does not align with the title and I have following suggestions for them.
1. Abstract: We emphasizes, it should be emphasize.
2. Introduction is not sufficient, significantly short and lacks the background for the need of current review (what is known, what this article explain etc.) Furthermore, the introduction lacks appropriate references.
3. section 2. Authors have directly jumped to the autonomic nervous system without discussing CNS, PNS. Also, they have not defined ANS and its components. This section also lacks supporting references. Kindly check following statement: The majority of neurons in the mouse celiac ganglion are tyrosine hydroxylase sympathetic and Neuropeptide Y positive neurons.
4. Kindly provide the source of figure 1. Also, Explain, whether figure two is published or unpublished. If it is previously published please provide permission to republish. Please remember that it is a review article so unpublished experimental figure will not be considered here.
5. Neural regulation of the immune system section is incomplete. It should be either well discussed or removed.
6. Overall section are not well organized and discussed, and lack appropriate citations/references.
Comments on the Quality of English LanguageEnglish language needs a check at some places. Longer sentences should be shorten.
Reviewer 2 Report
Comments and Suggestions for Authors
Dear Authers
Based on the comprehensive review of the manuscript "The Neuro-Immune Axis and its Therapeutic Potential for Primary Liver Cancer," here is a structured negative feedback focusing on key areas for improvement:
-
Clarity and Coherence: The manuscript occasionally suffers from a lack of clarity in its argumentation and the linkage between the neuro-immune axis and liver cancer progression. Simplifying complex sentences and improving the logical flow can enhance reader comprehension.
-
Depth of Literature Review: While the manuscript covers recent advancements, it lacks a critical analysis of conflicting studies. Including a broader spectrum of literature, especially studies with opposing viewpoints, would strengthen the review.
-
Methodological Details: The discussion on neurolysis and its potential benefits in HCC treatment is intriguing but lacks detailed exploration of methodology, including how these interventions could be systematically applied in clinical settings.
-
Statistical Analysis and Evidence: The manuscript references several studies but often does not delve into the statistical significance of the findings or their practical implications. A more detailed examination of the evidence base would lend more credibility to the conclusions drawn.
-
Innovation and Novelty: The review synthesizes existing knowledge but falls short in identifying new avenues of research or proposing novel hypotheses for the interaction between the neuro-immune axis and liver cancer. Encouraging more speculative or forward-looking thoughts could make the paper more impactful.
-
Figures and Illustrations: Some figures, while informative, could be improved for better visualization of the concepts discussed. High-quality, detailed diagrams or flowcharts summarizing the interactions within the neuro-immune axis could significantly enhance understanding.
-
Technical Accuracy: There are instances where the terminology used could be more precise or where more detailed explanations of specific mechanisms would be beneficial. Ensuring technical accuracy in describing neuro-immune interactions is crucial for the credibility of the review.
-
Conclusion and Implications: The conclusion provides a summary but lacks a strong statement on the future direction of research or potential clinical applications. A more definitive stance on the implications of the neuro-immune axis in liver cancer treatment would provide a clear takeaway for readers.
-
References: Some references appear to be outdated or not directly relevant to the manuscript's core thesis. Updating these references to include the most recent and pertinent literature would enhance the manuscript's relevance and authority.
-
General Formatting and Presentation: Attention to formatting details, including consistent citation styles and the organization of sections, would improve the manuscript's professional appearance and readability.
In summary, while the manuscript presents a valuable synthesis of the neuro-immune axis's role in liver cancer, addressing these areas for improvement could significantly enhance its contribution to the field.
Comments on the Quality of English LanguageMinor editing of English language required
Round 2
Reviewer 1 Report
Comments and Suggestions for Authors
The article still lacks to provide the information and has not been updated as per my viewpoint.
Comments on the Quality of English LanguageIts OK
Author Response
Thanks for your kind review.
Reviewer 2 Report
Comments and Suggestions for Authors
All the comments have been addressed
Author Response
Thanks for your kind review.